# Heat Transport on Ultrashort Time and Space Scales in Nanosized Systems: Diffusive or Wave-like?

**DOI:** 10.3390/ma15124287

**Published:** 2022-06-17

**Authors:** S. L. Sobolev, Weizhong Dai

**Affiliations:** 1Institute of Problems of Chemical Physics, Academy of Sciences of Russia, 142432 Chernogolovka, Moscow Region, Russia; 2Mathematics & Statistics, College of Engineering & Science, Louisiana Tech University, Ruston, LA 71272, USA; dai@coes.latech.edu

**Keywords:** heat waves, non-Fourier heat conduction, two temperature model, coupling, hyperbolic heat conduction

## Abstract

The non-Fourier effects, such as wave-like temperature propagation and boundary temperature jumps, arise in nanosized systems due to the multiple time and space scales nature of out-of-equilibrium heat transport. The relaxation to equilibrium occurs in successive time and space scales due to couplings between different excitations, whose relaxation times have different physical meanings and may differ significantly in magnitude. The out-of-equilibrium temperature evolution is described by a hierarchy of partial differential equations of a higher order, which includes both the diffusive and wave modes of heat transport. The critical conditions of transition from wave to diffusive modes are identified. We demonstrate that the answer to the question concerning which of these modes would be detected by experimental measurements may also depend on the accuracy of the experimental setup. Comparisons between the proposed approach and other non-Fourier models, such as the Guyer–Krumhansl and Jeffreys type, are carried out. The results presented here are expected to be useful for the theoretical and experimental treatment of non-Fourier effects and particularly heat wave phenomena in complex nanosized systems and metamaterials.

## 1. Introduction

In the last few years, there has been a growing community from different disciplines interested in the different aspects of out-of-equilibrium heat transport in nanosized systems, which occurs on ultrashort time and space scales [1,2,3,4,5,6,7,8,9,10,11]. The interest is largely motivated by technological needs such as thermal management in microelectronics and the ultra-fast laser processing of advanced metamaterials [1,4], i.e., artificial materials and media of designed properties, such as layered correlated materials [5,6]. Moreover, the heat transport at ultrashort space and time scales leads to unusual non-Fourier phenomena such as wave-like temperature propagation [4,5,6,12,13], size [14,15] and distance [15] dependent thermal conductivity, and boundary temperature jumps [1,14,15], which have raised an extensive body of literature concerning different conceptual questions of these phenomena [15,16,17,18,19,20,21,22,23,24,25,26,27,28,29,30]. The problem is that when the characteristic length of the process is of the order of the mean free path (MFP) of energy carriers and/or the characteristic time scale of the process is of the order of the mean free time (MFT) of energy carriers, the thermal dynamics occur under far from local equilibrium conductions and cannot be described by classical Fourier law based on the local equilibrium assumption [31]. In this case, one needs to use an approach which is not based on the local equilibrium assumption. The most simple and well-known generalization of the classical Fourier heat conduction equation of the parabolic type is the hyperbolic heat conduction equation (HHCE), which combines diffusive (dissipative) and wave modes of heat transport and transmits temperature discontinuity with a constant velocity without smoothing [12,25,26,29,30,31]. To our knowledge, the HHCE was first obtained under an assumption that the energy carriers move with a well-defined finite velocity (see discussion in [29]). While the classical heat conduction equation of the parabolic type results in the infinite propagation velocity of thermal disturbances, the HHCE removes this paradox leading to a finite value of propagation [12,22,25,31]. This property of the HHCE has attracted a wide range of interest mainly in the context of heat waves in liquid helium II (the so-called “second sound”) [12]. The wave-like energy propagation has also been detected in metals under ultrashort laser irradiation [32,33,34]. Recently, second sound has been observed in graphite at temperatures above 100 K [35]. The transfer equation of the hyperbolic type has also been used in the context of mass diffusion to study rapid alloy solidification under far from local equilibrium conditions [36,37].

However, the HHCE cannot be used to describe heat conduction in nanosized systems where the nonlocal space effects [14,15,16,17,18,21,22,26,27,28,31] and ballistic component of heat transport [14,15,16,17,19,23,24] lead to non-Fourier phenomena such as size-dependent thermal conductivity [1,9,14,15,16,17] and boundary temperature jumps [14,15,16,17,19], which can be observed even in the steady-state [14,15,16,17]. Note that the HHCE in the steady-state reduces to the classical heat conduction equation of the parabolic type, and consequently, cannot describe these effects. 

The nonlocal space effects [1,7,26,31] have been extensively studied in the context of heat conduction in nanosized systems [10,11,14,15,16,17,18,21,22,23,24,28], in metals under ultrashort (pico- and femtosecond) laser irradiation [4,31,32,33,34,38,39,40,41,42,43,44,45,46,47], and in biological [48,49,50] and heterogeneous systems [51,52,53]. Comprehensive reviews of different nonlocal heat conduction theories are presented in [41,42]. However, theoretical interpretations of these effects, as well as their experimental observations, lead to some controversies and misunderstandings (see [54,55,56,57,58,59] and the references therein). The same experimental results are treated by different authors either as diffusive or wave-like phenomena using different theoretical approaches [54,55,56,57,58,59]. In our view, the controversies mainly result from the fact that the role of the characteristics of the multiple time and space scales of heat transport in nanosized systems is not well understood [1,4,12,31]. In addition, all experimental measurements of transient temperature under far from local equilibrium conditions are subject to instrumental uncertainties, which, as it will be discussed in this paper, may be critical for the interpretation of the experimental data. Another question concerns the physical meaning of the relaxation times observed in the experiments and their identification on firm physical bases, which play important roles in constructing adequate theoretical models. Moreover, there are some uncertainties about the term “wave” itself in the context of heat transport. This paper is an effort towards a better understanding of the non-Fourier effects, particularly heat wave phenomena in nanosized systems, and the resolution of at least some of these controversies. Particular attention will be paid to the couplings between different excitations, which lead to the multi time and space nature of heat transport in nanosized systems and are of great importance from both theoretical and practical viewpoints for developing conceptually new nanodevices and metamaterials.

## 2. Non-Fourier Heat Conduction Models

### 2.1. Hyperbolic Heat Conduction Equation (HHCE)

Classical diffusive heat transport is described by Fourier law, which in 1D takes the form
$$q=-\lambda \frac{\partial T}{\partial x}$$
where *q* is heat flux, *T* is temperature, and λ is thermal conductivity. Combined with the energy conservation law, the Fourier law leads to the classical heat conduction equation of the parabolic type, which describes heat conduction as a pure thermal diffusion. This has the unphysical property that if a sudden change of temperature is made at some point on the body, it will be felt instantly everywhere, though with exponentially small amplitudes at distant points [12]. One may say that classical heat conduction theory gives rise to the infinite speed of the propagation of a temperature (thermal) wave [12]. This unpleasant physical property is also observed with other quantities, such as concentration [36,37] and viscous signals.

The most simple and well-known modification of the Fourier law, which overcomes the paradox of an infinite propagation velocity, is given by [12,29,30,31],
(1)$$q+\tau \frac{\partial q}{\partial t}=-\lambda \frac{\partial T}{\partial x}$$
where *τ* is the relaxation time to the local equilibrium. Equation (1) has been obtained from both macroscopic and microscopic approaches (see [12,16,17,18,22,25,26,29,30,31] and references therein). The values of the relaxation time τ range from tens of seconds in systems with heterogeneous inner structure and biosystems [49,50,51,52,53,54,55,56,57,58] to picoseconds in metals and dielectric solids [4,5,6,12,32,33,34,38,39,46,47]. Corresponding length scales range from a few millimeters to hundreds of nanometers. In particular, the modified Fourier law, Equation (1), can be treated as a first order expansion of the “time lag equation”:(2)$$q(t+\tau ,x)=-\lambda \frac{\partial T(t,x)}{\partial x}$$
which assumes that there is a time lag between the temperature gradient and the heat flux. Combining Equation (1) and the energy balance law, one obtains the HHCE in the form
(3)$$\frac{\partial T}{\partial t}+\tau \frac{\partial^{2}T}{\partial t^{2}}=k\frac{\partial^{2}T}{\partial x^{2}}+\bar{Q}+\tau \frac{\partial \bar{Q}}{\partial t}$$
where k=λ/C is thermal diffusivity, *C* is volumetric heat capacity, and *Q* is the heat source, $\bar{Q}=Q/C$. Equation (3) transmits temperature discontinuities without smoothing, which the amplitude exponentially decreases with time [12]. The propagation velocity of the temperature discontinuity is given by
(4)$$v_{T}={\left(k/\tau \right)}^{1/2}$$

Note that the HHCE is consistent with the second law of thermodynamics if the local nonequilibrium definition of entropy is used [30] (see also discussion in [18,56]).

### 2.2. Two-Temperature Models

#### 2.2.1. Two Temperature Parabolic Model (TTPM)

The two-temperature model (TTM) has been widely used for the analysis of out-of-equilibrium heat transport in metals upon laser excitation, where electrons and phonons are described as two subsystems with their own temperatures and interact through a coupling factor *α* [4,31,38,39,40,41,42,43,44,45,46,47]. The first version of the TTM consists of two coupled heat conduction equations for electrons and phonons:(5)$$C_{1}\frac{\partial T_{1}}{\partial t}=\lambda_{1}\frac{\partial^{2}T_{1}}{\partial x^{2}}+Q+\alpha (T_{2}-T_{1})$$
(6)$$C_{2}\frac{\partial T_{2}}{\partial t}=\alpha (T_{1}-T_{2})$$
where $C_{1}$ and $C_{2}$ are the volumetric heat capacities of the electrons and the lattice, respectively, *Q* is the heat source due to the laser irradiation, $\lambda_{1}$ is the thermal conductivity of electrons, α and is the coupling factor. The idea behind the introduction of their own temperatures for different subsystems is that the coupling between them arises at times that are long relative to the relaxation time of the fastest mode (for example, non-thermalized electrons in metals under ultrashort laser irradiation [4]), but short on the time scales characterizing the relaxation of the system to local equilibrium. Moreover, the TTM, demonstrated in Equations (5) and (6), assumes that the energy of a laser pulse is first absorbed by the electrons, which thermalize rapidly through electron–electron scattering. The electrons then transfer energy to the bulk material by electronic thermal diffusion and to the lattice through electron–phonon coupling, whereas heat diffusion through the lattice is neglected due to the relatively small thermal conductivity.

After some algebra, Equations (5) and (6) yield [4,31,38,39]
(7)$$\frac{\partial T_{1}}{\partial t}+\tau_{1,2}\frac{\partial^{2}T_{1}}{\partial t^{2}}=k_{1}\frac{\partial^{2}T_{1}}{\partial x^{2}}+\ell^{2}\frac{\partial^{3}T_{1}}{\partial t\partial x^{2}}+\bar{Q}+\tau_{1,2}\sigma \frac{\partial \bar{Q}}{\partial t}$$
(8)$$\frac{\partial T_{2}}{\partial t}+\tau_{1,2}\frac{\partial^{2}T_{2}}{\partial t^{2}}=k_{1}\frac{\partial^{2}T_{2}}{\partial x^{2}}+\ell^{2}\frac{\partial^{3}T_{1}}{\partial t\partial x^{2}}+\bar{Q}$$
where
(9)$$\tau_{1,2}=C_{1}C_{2}/\alpha (C_{1}+C_{2})$$
is the characteristic time of the energy exchange between the subsystems (coupling time),$\ell ={\left(\lambda_{1}C_{2}/\alpha (C_{1}+C_{2})\right)}^{1/2}={\left(\tau_{1,2}{\bar{k}}_{1}\right)}^{1/2}$ is the characteristic length, $k_{1}=\lambda_{1}/(C_{1}+C_{2})$ and ${\bar{k}}_{1}=\lambda_{1}/C_{1}$ are the normalized thermal conductivities, and $\sigma =(C_{1}+C_{2})/C_{1}$, $\bar{Q}=Q/(C_{1}+C_{2})$. As expected, Equations (7) and (8) are of the parabolic type with an infinite propagation velocity of thermal disturbances. It should be stressed that the TTM, which consists of two coupled *local* equations (see Equations (5) and (6)), leads to Equations (7) and (8), which are *nonlocal* both in time and space. The time scale of nonlocality is the characteristic time of the energy exchange between the subsystems $\tau_{1,2}$, while the space scale of the nonlocality ℓ is the length of the layer heated up by thermal diffusion during time $\tau_{1,2}$ [31,38]. Introducing the characteristic time scales for each subsystem as $t_{i}^{}=C_{i}/\alpha$, we can represent the characteristic time of the energy exchange between the subsystems (relaxation time) $\tau_{1,2}$ in the form of the Matthiessen rule as $\tau_{1,2}^{-1}=t_{1}^{-1}+t_{2}^{-1}$.

Thus, the coupling, i.e., the energy exchange between the subsystems, leads to the nonlocal effects, which play an important role at the time scale $t\propto \tau_{1,2}$ and at the space scale L∝ℓ. At a longer time scale $t>\tau_{1,2}$, the energy exchange between the subsystems tends to zero and $T_{1}=T_{2}$, while Equations (7) and (8) reduce to the classical Fourier heat conduction equations.

#### 2.2.2. Two-Temperature Hyperbolic Model (TTHM)

As it has been discussed above, the TTM, Equations (5) and (6), assumes that the interacting subsystems are each in separate local equilibria at all times and reach global equilibrium by exchanging energy and heat diffusion. In other words, it is assumed that, after excitation, both subsystems thermalize instantaneously, i.e., each of the two subsystems is in internal equilibrium. Thus, the validity range of the commonly used TTM is limited to times longer than the relaxation time to the local equilibrium of each subsystem.

To consider the case where one of the subsystems is far from the local equilibrium, for example, due to ultrafast laser excitation, the two-temperature hyperbolic model (TTHM) has been suggested [4,31,38,39]:(10)$$q_{1}+\tau_{1}\frac{\partial q_{1}}{\partial t}=-\lambda_{1}\frac{\partial T_{1}}{\partial x}$$
(11)$$C_{1}\frac{\partial T_{1}}{\partial t}=-\frac{\partial q_{1}}{\partial x}+Q+\alpha (T_{2}-T_{1})$$
(12)$$C_{2}\frac{\partial T_{2}}{\partial t}=\alpha (T_{1}-T_{2})$$
where $\tau_{1}$ is the relaxation time to the local equilibrium of subsystem 1, which is of the order of MFT of energy carriers, and $q_{1}$ is the heat flux in subsystem 1. Instead of the classical Fourier law commonly used by the TTM, the TTHM, Equations (10)–(12), employs the modified Fourier law, Equation (1), for the heat flux $q_{1}$ in the subsystem 1 (see Equation (10)), which takes into account the relaxation to the local equilibrium of subsystem 1 with the characteristic time $\tau_{1}$. Equation (11) basically states the conservation of energy density per unit time for subsystem 1 and takes into account that the internal energy changes due to the heat flux $q_{1}$, the external heat source *Q*, and the energy exchange with subsystem 2 described by the coupling constant α. Equation (12) is the energy conservation law for subsystem 2, which assumes that the energy of subsystem 2 changes only due to coupling with subsystem 1.

Combining Equations (10)–(12), we obtain [4,31,38,39]
(13)$$\frac{\partial T_{1}}{\partial t}+(\tau_{1}+\tau_{1,2})\frac{\partial^{2}T_{1}}{\partial t^{2}}+\tau_{1}\tau_{1,2}\frac{\partial^{3}T_{1}}{\partial t^{3}}=k_{1}\frac{\partial^{2}T_{1}}{\partial x^{2}}+\tau_{1,2}{\bar{k}}_{1}\frac{\partial^{3}T_{1}}{\partial t\partial x^{2}}+{\tilde{L}}_{1}[Q]$$
where ${\tilde{L}}_{1}={\left(C_{1}+C_{2}\right)}^{-1}\left[1+(\tau_{1}+\sigma \tau_{1,2})\partial /\partial t+\tau_{1}\tau_{1,2}\partial^{2}/t^{2}\right]$ is the differential operator. Note that the temperature of subsystem 2 is governed by the similar equation except for the differential operator, which is given by ${\tilde{L}}_{2}={\left(C_{1}+C_{2}\right)}^{-1}(1+\tau_{1}\partial /\partial t)$. Equation (13) contains two characteristic time scales, namely, $\tau_{1,2}$ being the characteristic time of the energy exchange between interacting subsystems (coupling time), and $\tau_{1}$ being the relaxation time to the local equilibrium of subsystem 1. The hierarchy of relaxation times $\tau_{1,2}$>$\tau_{1}$ implies that relaxation to equilibrium occurs in two stages: the first one is the relaxation to the local equilibrium of subsystem 1 due to the scattering of energy carriers inside subsystem 1 without interaction with subsystem 2, and the second one is the energy exchange between the subsystems, i.e., coupling. The stages are described by the corresponding hierarchy of evolution equations. On the shortest time scale *t*∼$\tau_{1}$, Equation (13) is of the hyperbolic type and transmits temperature discontinuities without smoothing at a finite velocity
(14)$$V_{1}={\left(\lambda_{1}/C_{1}\tau_{1}\right)}^{1/2}$$

On the intermediate time scale $t\propto \tau_{1,2}>\tau_{1}$, the coupling plays the most important role. In this case the TTHM, Equation (13), reduces to Equation (7), which is nonlocal both in time and space due to the coupling effects. On the longest time scale $t\gg \tau_{1,2}\gg \tau_{1}$, the coupling tends to zero and TTHM reduces to the classical Fourier heat conduction equation with a purely diffusive-like temperature evolution.

### 2.3. Guyer and Krumhansl (G–K) Equation

Guyer and Krumhansl [60] have solved the linearized Boltzmann equation with two relaxation times for the pure phonon field in terms of the normal-process collision operator and obtained the constitutive equation for the heat flux, which in 1D takes the form
(15)$$q+\tau_{R}\frac{\partial q}{\partial t}=-\lambda \frac{\partial T}{\partial x}+l^{2}\frac{\partial^{2}q}{\partial x^{2}}$$
where $\tau_{R}$ is relaxation time for momentum-non-conserving processes (the umklapp processes in which momentum is lost from the phonon system), $l^{2}=3\tau_{R}\tau_{N}c^{2}/5$, $\tau_{N}$ is the relaxation time for normal processes that preserve phonon momentum, *c* is the average (sound) speed of the phonons, and $\lambda =\tau_{R}c^{2}/3$ is thermal conductivity.

Combining the energy balance law and the G–K constitutive equation for heat flux, Equation (15), one obtains 


(16)
$$\frac{\partial T}{\partial t}+\tau_{R}\frac{\partial^{2}T}{\partial t^{2}}=k\frac{\partial^{2}T}{\partial x^{2}}+l^{2}\frac{\partial^{3}T}{\partial t\partial x^{2}}+\bar{Q}+\tau_{R}\frac{\partial \bar{Q}}{\partial t}-l^{2}\frac{\partial^{2}\bar{Q}}{\partial x^{2}}$$


Note that Equation (16), as well as the classical Fourier heat conduction equation, Equation (3), is of the parabolic type and transmits temperature disturbances with an infinite velocity.

It should be stressed that the G–K equation has been obtained for the pure phonon field in dielectric crystals at low temperatures, whereas electronic and some other interactions have been neglected [12,61]. Consequently, the analysis of the non-Fourier effects in systems with other heat conduction mechanisms, such as metals, biological, and heterogeneous systems [48,50,52], which are based on the use of the G–K equation, seems to be questionable from the methodological point of view. However, the structure of the G-K Equation (16) is similar to Equation (7), which is obtained using the TTM. The only difference is the absence of term $\partial^{2}Q/\partial x^{2}$ in Equation (7). This implies that both the TTM, Equation (7), and the G–K model, Equation (16), lead to similar temperature evolutions in systems without internal heat sources if the same boundary conditions are specified. The G–K equation seems to provide an adequate solution for the temperature distributions in heterogeneous systems [52,53] if only the physical meanings of the relaxation times are specified on firm physical bases.

Note that the second order space derivative of the flux also arises in the constitutive equation in the context of a mass transfer problem due to coupling between the stress and diffusion fields [62]. The G–K equation and its analogues are discussed with greater detail in [12,63,64]. 

### 2.4. Jeffreys Type Equation

Another nonlocal modification of the classical Fourier law is the heat-flux equation of the Jeffrey’s type [12]
(17)$$q+\tau_{J}\frac{\partial q}{\partial t}=-\lambda \frac{\partial T}{\partial x}-\tau_{J}\lambda_{eff}\frac{\partial^{2}T}{\partial t\partial x}$$
where $\lambda_{eff}$ is effective thermal conductivity and $\tau_{J}$ is relaxation time. Equation (17) is obtained by analogy with the equation for the stress and strain rate in liquids [12]. Combining the heat-flux equation of the Jeffreys type with the energy conservation law, one obtains the heat conduction equation of the Jeffreys type as follows
(18)$$\frac{\partial T}{\partial t}+\tau_{J}\frac{\partial^{2}T}{\partial t^{2}}=k\frac{\partial^{2}T}{\partial x^{2}}+\tau_{J}k_{eff}\frac{\partial^{3}T}{\partial t\partial x^{2}}+\bar{Q}+\tau_{J}\frac{\partial \bar{Q}}{\partial t}$$
where $k_{eff}=\lambda_{eff}/C$. The presence of an effective conductivity in the theory of heat conduction leads to exactly the same conceptual problem as pure diffusion—an infinite velocity of the propagation of a temperature wave. An equation of the Jeffreys type, Equation (18), looks similar to Equation (7) in the TTM and to the G–K Equation (16) (except for the last term $\partial^{2}Q/\partial x^{2}$ in Equation (16)).

## 3. Result and Discussion

### 3.1. Generalization of the G–K and the Jeffreys Type Equations

As it has been mentioned above, the G–K equation, Equation (16), and the Jeffreys type equation, Equation (18), are of the parabolic type and give rise to an infinite speed of propagation of the temperature wave. To eliminate this disadvantage, we can modify the corresponding constitutive equation for heat fluxes, keeping in mind that the left-hand side of Equations (15) and (17) are the first order approximations of the “time lag equation” for the heat flux $q(t+\tau_{i})$. Using a second order approximation of the lagged heat flux equation, we obtain the modified G–K and Jeffreys type constitutive equations for the fluxes as follows
(19)$$q+\tau_{R}\frac{\partial q}{\partial t}+\frac{\tau_{R}^{2}}{2}\frac{\partial^{2}q}{\partial t^{2}}=-\lambda \frac{\partial T}{\partial x}+l^{2}\frac{\partial^{2}q}{\partial x^{2}}$$
(20)$$q+\tau_{J}\frac{\partial q}{\partial t}+\frac{\tau_{J}^{2}}{2}\frac{\partial^{2}q}{\partial t^{2}}=-\lambda \frac{\partial T}{\partial x}-\tau_{J}\lambda_{eff}\frac{\partial^{2}T}{\partial t\partial x}$$

Combining Equations (19) and (20) with the energy conservation law, we obtain the generalized G–K and Jeffreys type heat conduction equations in the form
(21)$$\frac{\partial T}{\partial t}+\tau_{R}\frac{\partial^{2}T}{\partial t^{2}}+\frac{\tau_{R}^{2}}{2}\frac{\partial^{3}T}{\partial t^{3}}=k\frac{\partial^{2}T}{\partial x^{2}}+l^{2}\frac{\partial^{3}T}{\partial t\partial x^{2}}+\bar{Q}+\tau_{R}\frac{\partial \bar{Q}}{\partial t}+\frac{\tau_{R}^{2}}{2}\frac{\partial^{2}\bar{Q}}{\partial t^{2}}-l^{2}\frac{\partial^{2}\bar{Q}}{\partial x^{2}}$$
(22)$$\frac{\partial T}{\partial t}+\tau_{J}\frac{\partial^{2}T}{\partial t^{2}}+\frac{\tau_{J}^{2}}{2}\frac{\partial^{3}T}{\partial t^{3}}=k\frac{\partial^{2}T}{\partial x^{2}}+\tau_{J}k_{eff}\frac{\partial^{3}T}{\partial t\partial x^{2}}+\bar{Q}+\tau_{J}\frac{\partial \bar{Q}}{\partial t}+\frac{\tau_{J}^{2}}{2}\frac{\partial^{2}\bar{Q}}{\partial t^{2}}$$

The generalized G–K and Jeffreys type heat conduction equations, Equations (21) and (22), are of the hyperbolic type due to the additional time derivatives of the temperature $\partial^{3}T/\partial t^{3}$. The hyperbolicity of these equations implies that their solutions represent temperature discontinuities propagating without smoothing with finite velocities $V_{GK}=\sqrt{2}l/\tau_{R}=c\sqrt{6\tau_{N}/5\tau_{R}}$ and $V_{J}=\sqrt{2k_{eff}/\tau_{J}}$, which is usually treated as a heat (temperature) wave-like phenomenon. Note that Equations (21) and (22) are similar to Equation (13) obtained in the framework of the HHTM, Equations (10)–(12).

### 3.2. Hyperbolic Temperature Waves—Virtual Experiment

As it has been described above, the heat conduction equations of the hyperbolic type such as the HHCE, Equation (3); the TTHM, Equation (13); the generalized G–K equation, Equation (21); and the Jeffreys type heat conduction equations, Equation (22), transmit temperature discontinuity without smoothing at a finite but perhaps high velocity. However, the amplitude of the propagating temperature discontinuity decreases exponentially with distance [4,13], which may cause certain difficulties in the experimental detection of this wave. To illustrate the problem, let us consider the transient temperature distribution in a semi-infinite 1D heat conductor, *x* > 0, after a sudden change of temperature at *x* = 0 due to an ultrashort heat pulse. The amplitude of the temperature discontinuity $T_{f}$, which propagates with a constant velocity, Equation (4), exponentially decreases with time as it has been demonstrated in [12]
(23)$$\theta =(T_{f}-T_{0})/(T_{W}-T_{0})\approx \exp(-v_{T}^{2}t/2k)$$
where *θ* is the nondimensional amplitude of the temperature discontinuity, *T*_0_ is the initial temperature, $T_{W}=T(0)>T_{0}$ and is the temperature at the boundary *x* = 0 for *t* > 0. Considering that the distance between *x* = 0 and the temperature discontinuity front is given by $L=v_{T}t$, the exponent in Equation (23) can be represented as a function of distance as follows: $\exp(-v_{T}L/2k)$.

As the amplitude of the temperature discontinuity exponentially decreases with time, Equation (23), (or distance), it is clear that the discontinuity front cannot be detected experimentally at a long distance from *x* = 0 where the amplitude is very small. Let us denote the minimum amplitude of the temperature discontinuity that can be detected by the experimental setup as $\theta_{m}$ and virtually measure the temperature distribution in the heat pulse experiment. The temperature profiles obtained from the HHCE at different moments in times are shown in Figure 1.

As long as the temperature of the wave front $\theta (L,L/v_{T})$ exceeds the minimum temperature that can be detected by the experimental setup $\theta_{m}$, i.e., if $\theta (L,L/v_{T})>\theta_{m}$ (see black curve in Figure 1), the experimental setup detects the temperature change at the point *x =*
*L* at the moment in time $t_{ex}=L/v_{T}$. The dependence $t_{ex}\propto L$ implies a wave-like temperature propagation (see solid blue line in Figure 2).

If $\theta (L,L/v_{T})<\theta_{m}$ (see blue curve in Figure 1), the experimental setup is not able to detect the amplitude of the temperature discontinuity, which arrives from x=0 to x=L at the moment in time $t_{ex}=L/v_{T}$. In such a case, the change in the temperature at the point x=0 can be measured at the point x=L only when θ(L,t) reaches $\theta_{m}$, which occurs at a moment in time $t>L/v_{T}$. We denote by Λ the maximum value of the distance *L* which satisfies the condition $\theta (\Lambda ,\Lambda /v_{T})=\theta_{m}$ (see the green curve in Figure 1). If L<Λ, then $\theta (L,L/v_{T})>\theta_{m}$ and the experimental measurements yield $t_{ex}\propto L$, which implies a wave-like temperature propagation with constant velocity. If L>Λ, the experimental setup cannot detect moving temperature discontinuity because $\theta (\Lambda ,\Lambda /v_{T})<\theta_{m}$, so it detects the change of temperature at a later moment in time $t_{ex}>L/v_{T}$ when the system at the point x=L is heated up to $\theta_{m}$. Using Equation (23), we obtain Λ as a function of $\theta_{m}$ as follows
(24)$$\Lambda =-(2k/v_{T})\ln\theta_{m}$$

When L>Λ, the temperature distribution near the point x=L can be approximated by the solution of the parabolic heat conduction equation, which is well-known and given by
(25)$$\theta (x,t)=erfc(x/2\sqrt{kt})$$

Equation (25) can be represented as follows
(26)$$\theta (x,t)\approx 1-\frac{x}{\sqrt{\pi kt}}\left[1-\frac{1}{1!3}{\left(\frac{x}{2\sqrt{kt}}\right)}^{2}+\frac{1}{2!5}{\left(\frac{x}{2\sqrt{kt}}\right)}^{4}-\ldots \right]$$

Using Equation (26), the temperature at the point x=L for L>Λ at the moment in time $t>L/v_{T}$ can be approximated by
(27)$$\theta (x,t)\approx 1-\frac{L}{\sqrt{\pi kt}}$$

Considering that the experimental setup detects a minimum temperature $\theta_{m}$, Equation (27) gives
(28)$$\theta_{m}\approx 1-\frac{L}{\sqrt{\pi kt_{ex}}}$$

Using Equation (28), we obtain
(29)$$t_{ex}\approx \frac{L^{2}}{\pi k{\left(1-\theta_{m}\right)}^{2}}$$

In this case Equation (29) yields $t_{ex}\propto L^{2}$, which implies a diffusion-like temperature distribution rather than a wave-like one. The situation is illustrated in Figure 2, which shows $t_{ex}$ as a function of *L*. While L<Λ, the amplitude of the temperature discontinuity *θ* at the point *x =*
*L* exceeds $\theta_{m}$ (see the black curve in Figure 1) and the experimental measurements are able to detect the arrival of the temperature discontinuity, which implies a wave-like behavior of the temperature distribution with $t_{ex}\propto L$ (see the solid blue line at L<Λ in Figure 2), rather than a diffusive-like distribution (see dashed black curve at L<Λ in Figure 2). As soon as *L* exceeds Λ, the amplitude of the temperature discontinuity at the point *x = L* is small in comparison with $\theta_{m}$, and hence, the experimental setup is not able to detect the arrival of the temperature discontinuity to this point (see the blue curve in Figure 1). In this case, the temperature change at *x =* 0 will be detected at the point *x = L* at a later moment in time $t_{ex}>L/v_{T}$ as soon as the system at the point *x = L* is heated to $\theta =\theta_{m}$. Equation (29) implies that in such a case $t_{ex}\propto L^{2}$, which corresponds to a diffusion-like temperature distribution (see the solid blue curve at *x >* Λ in Figure 2), rather than a wave-like one (see the black dash-dotted line at *x >* Λ in Figure 2).

Now let us consider a virtual experiment when the distance *L* is fixed, whereas the minimum temperature amplitude *θ_m_* which can be detected by the experimental setup is variable. Equation (27) provides the amplitude of the temperature discontinuity arrived at the point *x = L* as follows
$$\bar{\theta }=1-{\left(\frac{v_{T}L}{\pi k}\right)}^{1/2}$$

When $\theta_{m}<\bar{\theta }$, the experimental setup is able to detect the amplitude of the temperature discontinuity arriving at the point *x = L* at the time $t=L/v_{T}$, and the measurements will demonstrate a wave-like temperature propagation $t_{ex}\propto L$. However, if $\theta_{m}>\bar{\theta }$, the experimental setup is not able to detect the amplitude of the temperature discontinuity arriving at the point *x = L* at the time $t=L/v_{T}$. In such a case, the experimental setup will detect temperature change at the point *x = L* at a later moment in time $t_{ex}>L/v_{T}$ demonstrating a diffusive-like temperature distribution with $t_{ex}\propto L^{2}$ (see Equation (29)). Thus, the same experimental results can be treated as a wave-like or diffusive-like temperature propagation depending on the accuracy of experimental measurements.

### 3.3. Hierarchy of Heat Conduction Equations in Systems with Couplings

Equation (13) obtained from the TTHM in the case of the systems without heat sources (*Q* = 0) can be represented as follows [31,38,64]
(30)$$\frac{\partial T}{\partial t}+(\tau_{1}+\tau_{1,2})\frac{\partial^{2}T}{\partial t^{2}}+\tau_{1}\tau_{1,2}\frac{\partial^{3}T}{\partial t^{3}}=k\frac{\partial^{2}T}{\partial x^{2}}+\ell^{2}\frac{\partial^{3}T_{1}}{\partial t\partial x^{2}}$$

Note that when $\tau_{1}$ and $\tau_{1,2}$ are identified with $\tau_{R}$, Equation (30) provides the generalized G–K equation, Equation (21), whereas when $\tau_{1}$ and $\tau_{1,2}$ are identified with $\tau_{J}$ Equation (30) yields the generalized equation of the Jeffreys type, Equation (22), where $\ell ={\left(\tau_{J}k_{eff}\right)}^{1/2}$.

Equation (30) takes into account the characteristics of the multiple time and space scales of heat transport in complex systems when a coupling between different subsystems occurs at time $\tau_{1,2}$ that is long relative to the relaxation time of the fastest mode due to the ballistic motion of energy carriers $\tau_{1}$, but short on the time scales characterizing the relaxation of the system to the local equilibrium $t_{eq}\propto L^{2}/k$, where *L* is the size of the system. The hierarchy of the time scale $\tau_{1}<\tau_{1,2}<t_{eq}$ leads to a corresponding hierarchy of evolution equations for the temperature.

On the shortest time scale $t\propto \tau_{1}$, the hyperbolic Equation (30) transmits a temperature wave as a discontinuity without smoothing at a finite velocity $v_{T}={\left(\ell^{2}/\tau_{1,2}\tau_{1}\right)}^{1/2}$. The discontinuity propagates along subsystem 1 without interaction with subsystem 2 because $\tau_{1,2}>\tau_{1}$. Considering that $\ell^{2}={\bar{k}}_{1}\tau_{1,2}$ in the TTM, we obtain Equation (14).

On the intermediate time scale $t_{eq}\propto \tau_{1,2}>\tau_{1}$, Equation (30) reduces to
(31)$$\frac{\partial T}{\partial t}+(\tau_{1}+\tau_{1,2})\frac{\partial^{2}T}{\partial t^{2}}=k\frac{\partial^{2}T}{\partial x^{2}}+\ell^{2}\frac{\partial^{3}T_{1}}{\partial t\partial x^{2}}$$

Equation (31) looks similar to the G–K equation (see the Equation (16)) and the Jeffreys type equation (see the Equation (18)). The similarity arises due to the characteristics of the multiple time and space scales of the heat conduction in complex systems. Indeed, the idea behind the TTM is that the relaxation of the energy carriers and the coupling between excitations occur on different time scales. The concept of the effective thermal conductivity $\lambda_{eff}$ in the Jeffreys type model has been introduced for systems in which different substructures in a material relax at different rates, i.e., when slow modes and fast modes can be well-identified [12]. In other words, an effective thermal conductivity arises at times that are long relative to the relaxation times of the fast modes but short on the time scales characterizing the relaxation of the slow modes [12], which corresponds to the hierarchy of the time scales in the TTM.

On the short length scale ℓ→0, Equation (31) reduces to the HHCE, Equation (3), with the finite velocity of propagation of temperature discontinuity $V_{T}={\left(k/\tau_{1,2}\right)}^{1/2}$. However, if ℓ≠0, Equation (31) is of the parabolic type with an infinite velocity of propagation of temperature disturbances. Thus, the nonzero nonlocal effects ℓ≠0 smooth the discontinuity leading to an infinite propagation velocity of temperature distribution such as in pure diffusion. This type of smoothing is analogous to the smoothing by the viscosity of the shock waves in gas dynamics [12]. For very small but still nonzero values of l, Equation (31) predicts a continuous wave-like structure whose dimensionless front thickness δl is proportional to ${\left(\lambda_{eff}x/\lambda \right)}^{1/2}$, where x is a dimensionless coordinate [12] (see the Figure 3). Thus, small nonlocal effects may not destroy effective wave propagation with a continuous temperature distribution on small time and space scales. Moreover, if the minimum dimensionless length that can be detected in the experiment $l_{ex}$ exceeds δl, then the experimental measurements will demonstrate a propagation of a wave-like temperature discontinuity (see the black short-dashed line in Figure 3). As the distance *x* increases, the width of the wave-like front δl  increases and it transforms into the classical purely diffusive rather than wave-like temperature distribution. Thus, in the heat pulse experiment, a wave-like structure can be observed only on a short length scale, while on a long length scale the temperature distribution tends to be a purely diffusive one. Note that at the intermediate time scale $t\propto \tau_{1,2}>\tau_{1}$ the wave-like temperature propagation arises due to coupling effects, whereas on the shortest time scale $t\propto \tau_{1}$ it is due to the ballistic motion of the energy carriers with a well-defined group velocity.

Thus, the relaxation of complex systems from a highly nonequilibrium state to equilibrium is a multi-stage process which includes a relaxation to the local equilibrium of the energy carriers and couplings between different excitations occurring in successive time and space scales. This allows one to introduce their own temperatures for different subsystems and consider the relaxation to equilibrium as a multi-temperature process. Even a small difference in temperature between subsystems, even in low energy nanodevices, leads to relatively large temperature gradients due to the coupling at a short space scale, which implies that such nanodevices operate under local non-equilibrium conditions. The hierarchy of successive time scales due to relaxation to the local equilibrium of the energy carriers and coupling between different subsystems leads to a corresponding hierarchy of evolution equations for the temperature. The highest order partial differential equation of the hierarchy considers all the excitations and couplings, whereas the last equation in the hierarchy is nothing else but the classical Fourier heat conduction equation for a local equilibrium situation. The hierarchy can be used to study the out-of-equilibrium thermal dynamics of complex systems with couplings of different physical natures, such as electron–phonon coupling in metals [4,31,38,39,40,41,42,43,44,45,46,47]; phonon–phonon coupling in graphene and graphite [35,65]; coupling between different excitations in metamaterials, such as electron–electron coupling in layered correlated materials and 3D printed structures [5,6]; electron–phonon couplings in a 2D material [13], coupling between different subsystems in biological systems [48,49,50]; and systems with heterogeneous inner structures, such as foams and composites [8,51,52,53].

It should be noted that, strictly speaking, at ultrashort space and time scales, the continuum hypothesis breaks down and the usual description based on the partial differential equations is no longer valid. In this paper, we consider a relatively small deviation from equilibrium, i.e., the so-called “weak” nonlocality, which can be described in the framework of the continuum hypothesis. However, when the deviation from equilibrium is strong, namely, when the characteristic time and space scales of the process are close to or even less than the MFT and MFP of energy carriers, respectively, the continuum hypothesis is not valid anymore. An example of a heat conduction approach, which is not based on the continuum hypophysis, is provided by the discrete variable model [14,15,26,31,38]. The model assumes that space and time are discrete variables, which allows one to study heat conduction in nanosized systems under strong deviation from the equilibrium, for example, heat conduction across a nano film [14,15], rapid phase transportation [37], and heat conduction in layered correlated materials [5,6].

### 3.4. Comments on Definition of Temperature in Out-of-Equilibrium Systems

The non-Fourier phenomena in heat conduction, and temperature waves in particular, arise under far from local equilibrium conditions, while the temperature, strictly speaking, is defined only for the global (or local) equilibrium. This implies that the *nonequilibrium temperature* should be used to describe non-Fourier effects and heat wave propagation [30,61,66]. It has been demonstrated that in 1D heat conducting systems, the modified Fourier law, Equation (1), governs the direction and value of the heat flux by the gradient of the *kinetic temperature*, which is proportional to the local energy density, and such an evolution of the *kinetic temperature* is described by the HHCE, Equation (3) [30,61,66]. However, the “effective” or “entropic” temperature obtained based on the Shannon informational entropy and Gibbs equation characterizes the thermalized fraction of the internal energy density as being similar to the equilibrium thermodynamic temperature and allows one to extend the meaning of other equilibrium variables, such as the thermal conductivity and heat capacity, to the nonequilibrium scenario [30,61,66]. In an equilibrium, the kinetic and entropic temperatures coincide, whereas far from equilibrium, when the heat flux increases, they differ substantially. The ratio of the heat flux to its maximum possible value plays the role of an order parameter—it varies from zero in the equilibrium (disordered) state to unity in the nonequilibrium (ordered) state [30,61,66]. Note that the out-of-equilibrium heat transport can be described not only in terms of temperature, but also in terms of heat flux, for which the definition does not rely on the local equilibrium assumption [14,15,30,31]. While the HHCE, Equation (3), describes evolution of the kinetic temperature *T*, a similar hyperbolic equation describes evolution of the heat flux [30,31]:
$$\frac{\partial q}{\partial t}+\tau \frac{\partial^{2}q}{\partial t^{2}}=k\frac{\partial^{2}q}{\partial x^{2}}-k\frac{\partial \bar{Q}}{\partial x}$$


This equation implies that the heat flux evolution represents a discontinuity propagating without smoothing with a constant velocity, given by Equation (4).

## 4. Conclusions

The relaxation of complex systems from a highly nonequilibrium state to equilibrium is a multi-stage process, which generally includes the ballistic motion of energy carriers and coupling between different excitations occurring in successive time and space scales. The hierarchy of the scales allows us to consider the relaxation as a multi-temperature process, which is described by the corresponding hierarchy of evolution equations. The fastest mode in the hierarchy corresponds to the ballistic motion of the energy carriers, which under a single relaxation time approximation can be described by the HHCE. In such a case, an imposed temperature discontinuity propagates without smoothing at a finite velocity. However, the amplitude of the discontinuity exponentially decreases with distance, which may cause some difficulty in its experimental detection. The discontinuity is followed by a continuous wave-like structure, which arises due to couplings between different excitations. The structure can be experimentally detected at times that are long relative to the relaxation time of the fastest mode but short on the time scales characterizing the relaxation of the system to equilibrium. The critical conditions of the transition from the wave-like to diffusive-like modes are identified.

While the G–K model has been developed for pure phonon transport in dielectric crystals at low temperatures, its application for other systems, such as heterogeneous structures and biosystems, is questionable. The multi temperature approach seems to be more universal and can be used to study the non-Fourier heat phenomena in complex nanosized systems, such as metals [4,38,39,40,41,42,43,44,45,46]; heterogeneous systems [8,19,51,52,53]; metamaterials such as layered correlated materials [5,6], nanosized graphene [65], and bio-heat systems [48,49,50], where coupling between different excitations play a significant role.

## Figures and Tables

**Figure 1 materials-15-04287-f001:**
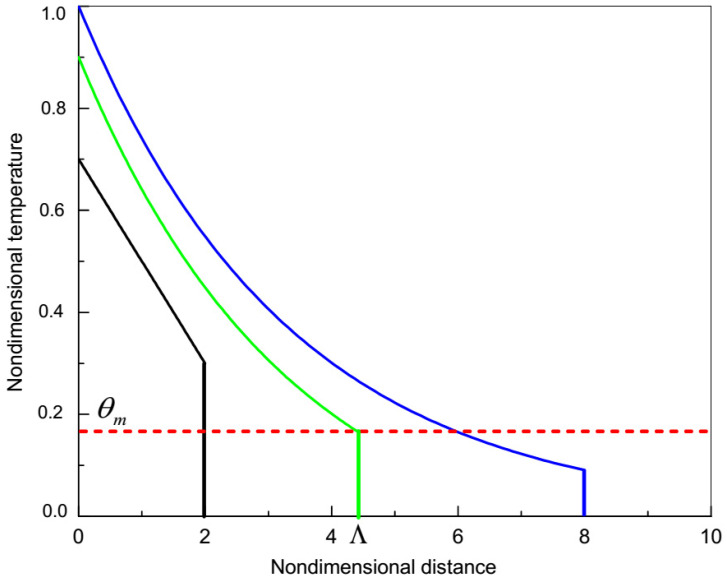
Nondimensional temperature distributions due to HHCE in successive nondimensional moments in time $t_{black}<t_{green}<t_{blue}$. Red dashed line—the minimum temperature detected in the experiment $\theta_{m}$.

**Figure 2 materials-15-04287-f002:**
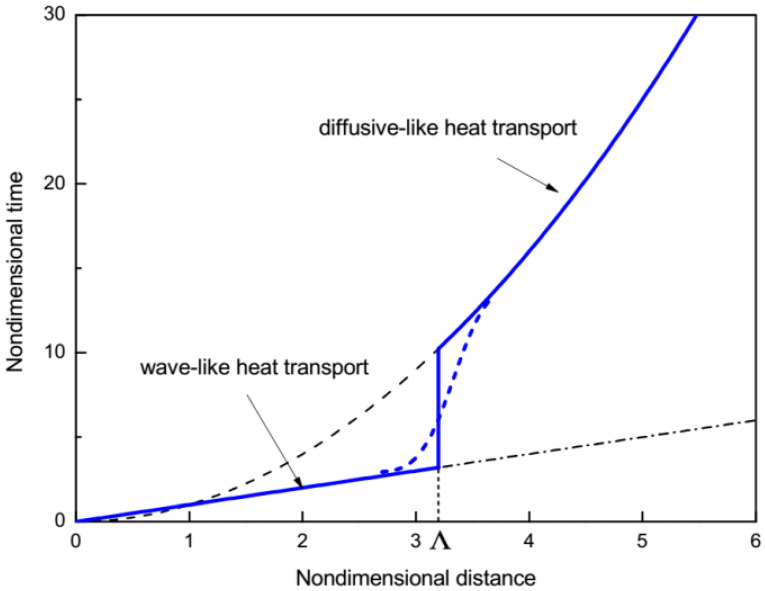
The nondimensional $t_{exper}\;$ as a function of the nondimensional distance *L* (blue curve). Black dash-dotted line—purely wave propagation $t_{exper}∼\;L$; black dashed—purely diffusive heat conduction $t_{exper}∼\;L^{2}$. At the critical point L=Λ occurs transition from the wave-like to diffusive-like mode (blue dashed curve—continuous transition from between the mode).

**Figure 3 materials-15-04287-f003:**
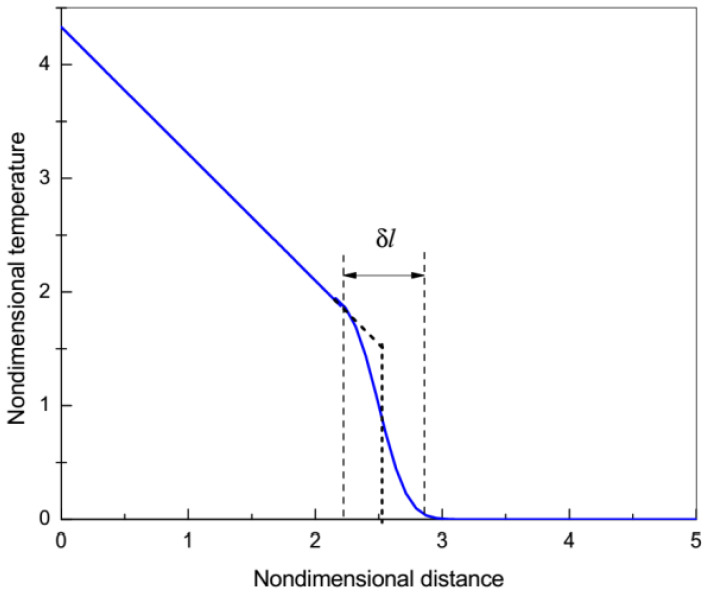
Schematic representation of the continuous wave-like structure of thickness *δl* arising due to coupling effects (blue curve). If the lab space scale exceeds *δl*, the front can be treated as a discontinuous one (black dashed curve).

## Data Availability

All data regarding the simulation and modeling are available on request.

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
