# Peer review of "Heat Transport on Ultrashort Time and Space Scales in Nanosized Systems: Diffusive or Wave-like?"

_materials, 2022, doi:10.3390/ma15124287_

Round 1

Reviewer 1 Report

In this paper    a  multi temperature  universal approach  is   developed,  that can be used to investigate the non-Fourier heat phenomena in nanosized systems, such as metals, heterogeneous systems,  metamaterials,  layered materials, nanosized graphene, bio-heat systems,  complex systems, where  the  relaxation from a highly non-equilibrium state to equilibrium is a multi-temperature  process,  including coupling between different excitations occurring in successive time and space scales.

The hierarchy of the scales  is described by a corresponding hierarchy of evolution equations.  The critical conditions of transition from the wave-like to diffusive-like modes are studied.  Comparisons among  the proposed approach and other non-Fourier models, such as the Guyer-Krumhansl and Jeffers-type, are given.

The paper is well written and organized.  The results are very interesting for their applications in several  fields of science. I suggest to publish this paper

but  it is  good that in the references the authors add the following articles:

V. A. Cimmelli, Different thermodynamic theories and different conduction laws,       J. Non-Equilib. Thermodyn. 2009 · Vol. 34 · pp. 299–333, where   different non-equilibrium thermodynamic  approaches  that study the heat conduction are presented,

P.  Ván, T. Fülöp, Universality in heat conduction theory: weakly nonlocal thermodynamics,  Ann. Phys. (Berlin) 524, No. 8, 470–478 (2012) / DOI 10.1002/andp.201200042 , where  an universal approach is developed in the framework of the non-equilibrium thermodynamics, from which the well-known Fourier, Maxwell–Cattaneo–Vernotte, Guyer–Krumhansl, Jeffreys-type, and other equations of heat conduction are obtained,  with the related temperature equations.

Furthermore, equation (1), row 102,  pag. 3,  is the celebrated Maxwell-Cattaneo-Vernotte (MCV) equation (see Maxwell 1872; Cattaneo 1948; Vernotte 1958;  Fichera, 1992)

Maxwell, J. C. (1872). Theory of Heat. Longman, London, UK. DOI: 10.1017/CBO9781139057943.

 Cattaneo, C. (1948). “Sulla conduzione del calore”. Atti del Seminario Matematico e Fisico dell’ Università di Modena 3, 83–101. DOI: 10.1007/978-3-642-11051-1_5.

Vernotte, P. (1958). “Les paradoxes de la théorie continue de l’équation de la chaleur”. Comptes Rendus 264(22), 3154–3155.

Fichera, G. (1992). “Is the Fourier theory of heat propagation paradoxical?” Rendiconti del Circolo Matematico di Palermo 41(1), 5–28. DOI: 10.1007/BF02844459.

Author Response

According to the reviewer comment I added the references to the list of references and added corresponding comment in the text.

“Comprehensive reviews of different nonlocal heat conduction theories are presented in [41,42]”. (lines 66,67)

[41] Cimmelli, V.A. Different thermodynamic theories and different conduction laws. J. Non-Equilib. Thermodyn. 2009, 34, 299–333.

 [42] Ván, P.; Fülöp, T. Universality in heat conduction theory: weakly nonlocal thermodynamics, Ann. Phys. (Berlin) 2012, 524, 470–478.  (2012).  DOI 10.1002/andp.201200042.

We agree with the Reviewer that Eq.(1) is the celebrated Maxwell-Cattaneo-Vernotte (MCV) equation (see Maxwell 1872; Cattaneo 1948; Vernotte 1958;  Fichera, 1992). However, these papers are well-known and cited in the reviewers in the list of references in the manuscript.  We wanted to attract attention of the scientific community to other less known, but, in our opinion, important papers on the subject, which by the way were published earlier than the Cattaneo paper (see discussion in [29-31]). It should be mentioned that the Cattaneo approach is criticized because Eq.(1) was obtained in his work due to incorrect mathematical procedure (see discussion in [12,29]).  

Reviewer 2 Report

This work gives a clear overview of some of the existing mathematical descriptions of non-Fourier heat transfer. However, there is a large number of articles and review papers that cover and compare the different non-Fourier models, which is why I would suggest to the authors to make a clear statement about why their work stands out from previous publications. 

Secondly, I think it would be interesting that the autors gave some information about the orders of magnitude of the different non-classical time- and length-scales. Without this, the results presented here may be too theoretical and not of practical use for experimental people. Moreover, there is a limit where continuum models cannot be applied as the medium cannot be assumed as such anymore. Should these parameters fall below this limit, then the equations may not be valid anymore.

Overall, the contents of this draft is interesting and worth publishing in this journal, provided the authors can address these two points above.

Author Response

  1. We agree with the reviewer that “there is a large number of articles and review papers that cover and compare the different non-Fourier models”. It is impossible to discuss all the papers on non-Fourier heat conduction in one manuscript and our main purpose was not to make a comprehensive review of all the non-Fourier models (some reviewers are cited in the list of references), so we tried to choose the most relevant ones and less known to the scientific community. Of course, the choice is subjective. We added in the list of references two reviews on the subject [41,42].
  2. According to the reviewer remark, we added “some information about the orders of magnitude of the different non-classical time- and length-scales”. The follows text is added to the manuscript – lines 106-109:

“The values of relaxation time τ range from tens of seconds in systems with heterogeneous inner structure and biosystems [49-58] to picoseconds in metals and dielectric solid [4-6,12,32-34,38,39,46,47]. Corresponding length scales range from a few millimeters to hundreds of nanometers “.

  1. I absolutely agree with the reviewer that “there is a limit where continuum models cannot be applied”. It is very important that the reviewer raises this question. I tried to discuss this problem in my early papers [26,31,38] but scientific community does not pay much attention to the problem. All the equations considered in the manuscript assume that the continuum hypothesis holds. We mentioned in the text (lines 56-58) that “However, the HHCE cannot be used to describe heat conduction in nanosized system where the space nonlocal effects [14-18,21,22,26-28,31] and ballistic component of heat transport [14-17,19,23,24] lead to the non-Fourier phenomena …”, which implies the limitation of the HHCE. Moreover, we added the following to the text (lines 443-445):

“It should be noted that, strictly speaking, on ultrashort space and time scales the continuum hypothesis breaks down and usual description based on the partial differential equations is not valid any more. In this paper we consider relatively small deviation from equilibrium, i.e. the so-called “weak” nonlocality, which can be described in the framework of the continuum hypothesis. However, when the deviation from equilibrium is strong, namely, when the characteristic time and space scales of the process are close to or even less than the MFT and MFP of energy carriers, respectively, the continuum hypothesis is not valid anymore.  An example of heat conduction approach, which is not based on the continuum hypophysis, is provided by the discrete variable model [14,15,26,31,38]. The model assumes that space and time are discrete variables, which allows one to study heat conduction in nanosized systems under strong deviation from equilibrium, for example, heat conduction across a nano film [14,15], rapid phase transportation [37], heat conduction in layered correlated materials [5,6]”.

Reviewer 3 Report

Please find attached a pdf file with my comments.

Thank you very much for your attention.

Best Regards.

Author Response

  1. We agree with the reviewer that the entropy principal is very important in local nonequilibrium heat conduction and needs special and careful consideration. In particular, the principal is under discussion in the papers cited in the manuscript [18,30,56,65,66]. We think that in this manuscript it is enough to make short comment on problem (lines 120-121)

“Note that the HHCE is consistent with the second law of thermodynamics if the local nonequilibrium definition of entropy is used [30,65,66] (see also discussion in [18,56]).”

In the future works the problem will be discussed in more detail.

We also corrected all misprints mentioned by the reviewer.